# Transcriptional changes in *Plasmodium falciparum* upon conditional knock down of mitochondrial ribosomal proteins RSM22 and L23

**Swati Dass, Michael W. Mather, Joanne M. Morrisey, Liqin Ling¤, Akhil B. Vaidya, Hangjun Ke** *

Center for Molecular Parasitology, Department of Microbiology and Immunology Drexel University College of Medicine, Philadelphia, Pennsylvania, United States of America

¤ Current address: Department of Laboratory Medicine, West China Hospital, Sichuan University, Chengdu, China

* hk84@drexel.edu

**Data Availability Statement:** The FASTq sequence files of PfRSM22 and PfMRPL23 from RNA sequencing reported in this paper are deposited in NCBI https://www.ncbi.nlm.nih.gov/sra/

## Abstract

The mitochondrion of malaria parasites is an attractive antimalarial drug target, which require mitoribosomes to translate genes encoded in the mitochondrial (mt) DNA. *Plasmodium* mitoribosomes are composed of highly fragmented ribosomal RNA (rRNA) encoded in the mtDNA. All mitoribosomal proteins (MRPs) and other assembly factors are encoded in the nuclear genome. Here, we have studied one putative assembly factor, RSM22 (Pf3D7_1027200) and one large subunit (LSU) MRP, L23 (Pf3D7_1239100) in *Plasmodium falciparum*. We show that both proteins localize to the mitochondrion. Conditional knock down (KD) of PfRSM22 or PfMRPL23 leads to reduced cytochrome $bc_1$ complex activity and increased sensitivity to $bc_1$ inhibitors such as atovaquone and ELQ-300. Using RNA sequencing as a tool, we reveal the transcriptomic changes of nuclear and mitochondrial genomes upon KD of these two proteins. In the early phase of KD, while most mt rRNAs and transcripts of putative MRPs were downregulated in the absence of PfRSM22, many mt rRNAs and several MRPs were upregulated after KD of PfMRPL23. The contrast effects in the early phase of KD likely suggests non-redundant roles of PfRSM22 and PfMRPL23 in the assembly of *P. falciparum* mitoribosomes. At the late time points of KD, loss of PfRSM22 and PfMRPL23 caused defects in many essential metabolic pathways and transcripts related to essential mitochondrial functions, leading to parasite death. In addition, we enlist mitochondrial proteins of unknown function that are likely novel *Plasmodium* MRPs based on their structural similarity to known MRPs as well as their expression profiles in KD parasites.

PRJNA801275. All additional data are contained within this manuscript and in the Supporting Information.

**Funding:** The work was supported by NIH/NIAID grants to Dr. Hangjun Ke (K22AI127702) and Dr. Akhil Vaidya (AI028398). The publication fee is sponsored by Library Pilot Fund of Drexel University College of Medicine.

**Competing interests:** The authors declare that they have no conflicts of interest with the contents of this article.

**Abbreviations:** The abbreviations used are, ; PfRSM22, Plasmodium falciparum mitochondrial ribosomal associated protein of small subunit; METTL17, methyltransferase like protein 17; SAM, S-adenosyl-L-methionine dependent methyltransferase; mitoribosomes, mitochondrial ribosome; MRP, mitoribosomal protein; mtETC, mitochondrial electron transport chain; mtDNA, mitochondrial DNA; cyt, cytochrome; cox1, cytochrome oxidase subunit I; cox3, cytochrome oxidase subunit III; TetR, tetracycline repressor; DOZI, development of zygote inhibited; cryo-EM, cryo-electron microscopy; gRNA, guider RNA; aTc, anhydrotetracycline; IDC, intraerythrocytic cycles; KEGG, Kyoto Encyclopedia of Genes and Genomes; GO, Gene Ontology; RPKM, Reads Per Kilobase Million; GSEA, Gene Set Enrichment Analysis.

## Introduction

The mitochondrion is an essential organelle in eukaryotes that emerged from an alpha proteo-bacterial endosymbiont ~ two billion years ago [1]. It has diversified tremendously with every evolving eukaryote adapting the mitochondrion's roles according to the physiological demands of each organism's evolutionary niche [2]. A vast literature is available on understanding the composition and physiology of mitochondria in model eukaryotic systems [3, 4]. This information has been exploited to help understand the evolutionary and functional differences in the mitochondria of less studied eukaryotes, including *Plasmodium spp.*, the causative agent of malaria, but much remains unknown about the function of organelles in these early diverging organisms [5, 6].

According to the latest World Malaria Report, there has been a spike of 14 million malaria cases in 2020, raising the death toll to 627,000 around the world [7]. This signifies malaria as a heightened global health burden and demands a thorough understanding of *Plasmodium* biology to support advances in antimalarial drug development. The *Plasmodium* mitochondrion has been validated as an antimalarial drug target due to its essentiality in the parasite and differences from the human counterpart. The parasite mitochondrial electron transport chain (mtETC) is a target of an array of clinical and pre-clinical antimalarials, such as atovaquone and ELQ-300, which do not affect the human mtETC at pharmacologically relevant concentrations [8, 9]. Despite its importance, our understanding of the *Plasmodium* mitochondrion remains limited [10]. The *Plasmodium* mitochondrial genome encodes only three mtETC proteins (*cyt b*, COX I, and COX III). Other than these three proteins, this genome encodes mt rRNA genes on both strands of the mtDNA [11, 12]. Unlike the rRNAs of most other organisms, the *Plasmodium* mt rRNAs are fragmented into 39 distinct molecules ranging in length from 20 to 200 bases. Another unusual phenomenon of *Plasmodium* mt rRNA is the presence of a short oligo (A) tail at the 3' end of most mt rRNA transcripts [13]. On the other hand, all MRPs are encoded in the parasite nuclear genome and post translationally transported to the parasite mitochondrion where they assemble along with mt rRNAs to form a functional mitoribosome [14].

Currently, 43 putative MRPs in *Plasmodium falciparum* (Pf) have been annotated based on their sequence similarity with known MRPs of bacterial and mitochondrial origin (www. PlasmoDB.org). However, the species-specific *Plasmodium* MRPs remain entirely unknown [15, 16]. We have previously studied four annotated MRPs in *P. falciparum*, including PfMRPL13, PfMRPS12, PfMRPS17, and PfMRPS18 [17, 18]. We have shown that knocking down PfMRPs reduces the cytochrome *c* reductase activity of the *bc₁* complex, consistent with an essential role of mitoribosomes in translating mtETC proteins [17]. We have also shown that disrupting *P. falciparum* mitoribosomes impairs mitochondrial metabolic pathways, rendering the parasite hypersensitive to antimalarial drugs targeting the mtETC [18]. However, it remains unknown how the loss of mitoribosomes affects the overall health of the parasite, gradually leading to parasite demise. Apart from the four MRPs mentioned above, the other 39 MRPs have not been characterized in *Plasmodium*.

Here, we have studied two annotated MRPs, PfRSM22 and PfMRPL23. We used CRISPR/ Cas9 [19] and the TetR-DOZI-aptamer system to generate conditional KD lines to control expression of PfRSM22 and PfMRPL23 and insert a triple hemagglutinin (3xHA) tag at the C-terminal end of the proteins. Using these transgenic parasite lines, we confirm that PfRSM22 and PfMRPL23 are localized to the mitochondrion and their genetic knock down leads to mitochondrial functional defects. To understand the global and mitochondrial transcriptomic changes upon KD of PfRSM22 or PfMRPL23, we have performed RNA sequencing of the asexual stage parasites at multiple time points. Here, we categorize the parasite morphological and

transcriptomic changes at the early and late phases of KD. In addition, we have identified differentially regulated *Plasmodium* proteins of unknown function that exhibited structural similarity to distant MRPs of other organisms. These proteins could represent new *Plasmodium* MRP candidates.

## Materials and methods

### Parasite culture

The D10 strain of *P. falciparum* was cultured in O$^+$ human RBCs (Interstate Blood Bank, TN) supplemented with RPMI 1640 medium along with 0.3–0.5% Albumax I (Invitrogen), 2g/L sodium bicarbonate, 10 mg/L hypoxanthine (Thermo Fisher Scientific), 15 mM HEPES (Millipore Sigma) and 50 mg/L gentamycin (VWR). Parasite culture was maintained at 37˚C with conditions previously described [17].

### Plasmid construction and transfection

The plasmid carrying the TetR-DOZI-aptamer elements for conditional KD (pMG75) was kindly provided by Dr. Jacquin Nile's group from MIT. Primers used for cloning the homologous regions of interest into the pMG75 vector are listed in S1 Table. The pMG75 vector was used to modify the genomic locus of PfRSM22 (Pf3D7_1027200). Briefly, the original pMG75noP-ATP4-8apt-3HA plasmid was linearized by AflII and BstEII, to remove the ATP4 inserts. PfRSM22 5'HR region was amplified from *P. falciparum* genomic DNA using primers (P2+P3) giving a PCR product of 868 bp. The 3'HR region was downstream of the stop codon and was amplified using primers (P4 + P5) giving a PCR product of 601 bp. The three pieces including the linearized vector, PfRSM22 5'HR and PfRSM22 3'HR were joined together using NEB HiFi DNA assembly master mix. The gRNAs of PfRSM22 were present at the end of 5'HR, where Cas9 would introduce a cut to create the expected modification in the transgenic parasites. To avoid repetitive cutting, synonymous mutations were introduced in the reverse primer of the 5'HR (P3) within the gRNA region. The modified pMG75 vector was sequenced using primers P11 and P12 to verify the 3'HR and 5'HR inserts, respectively.

The PfMRPL23 5'HR and 3'HR were cloned into a pMG75 derivative one after the other. Briefly, the pMG75 vector containing mt-DNA polymerase I (Pf3D7_0625300) was digested with BstEII and SacII to remove mt-DNA pol 3'HR. PfMRPL23 3'HR was amplified from genomic DNA using primers P16 +P17. The amplified 3'HR (1000 bp) was digested with SacII and BstEII and inserted into the digested pMG75 vector. Presence of the PfMRPL23 3'HR and absence of the mt-DNA pol 3'HR was confirmed by PCR followed by sequencing. The pMG75 vector carrying PfMRPL23 3'HR was digested with SacII and SalI to remove mtDNA pol 5'HR. PfMRPL23 5'HR was amplified using primers P14+P15. The amplified product (853 bp) was digested with SacII and SalI and inserted into the pMG75 vector carrying PfMRPL23 3'HR, resulting in the final plasmid pMG75-PfMRPL23-3HA-8apt. The modified pMG75 vector was sequenced using primers P11 and P12 to the verify 3'HR and 5'HR inserts, respectively.

Guide RNA (gRNAs) were designed using the Eukaryotic Pathogen CRISPR guide RNA Design Tool (http://grna.ctegd.uga.edu). High efficiency score gRNAs having no off-target matches were selected and PfRSM22 gRNAs (P7 and P9) were cloned in a No flag (NF)-Cas9-yDHOD (-) plasmid. Briefly, NF-Cas9-yDHOD (-) vector [18] was digested with EcoRI and joined with oligos P7 and P9 individually by NEB HiFiDNA assembly. Correct infusion of PfRSM22 gRNA1-NF-Cas9-yDHOD (-) was confirmed by PCR using primers P8+P23 whereas PfRSM22 gRNA2 insertion was confirmed by PCR using P10+P23. The inserted gRNAs were further confirmed by sequencing. PfMRPL23 gRNAs were cloned into the

M-Cas9-yDHOD (-) vector which still had a flag tag present in Cas9. The M-Cas9-yDHOD(-) vector digested with EcoRI was joined with oligos P19 and P21 individually by NEB HiFiDNA assembly. Fusion of gRNAs into M-Cas9-yDHOD was verified by PCR using primers P20 +P23 (gRNA1) or P22+P23 (gRNA2) and by sequencing.

All DNA fragments were amplified using high fidelity DNA polymerases (New England Biolabs®, Inc) and confirmed by Sanger sequencing (Genewiz LLC). Transformation of pMG75 related plasmids was performed using stable competent *E. coli* cells (New England Bio-labs®, Inc) and bacteria were grown at 30˚C to maintain the stability of 8X aptamer repeats. Transformation of Cas9 related plasmids was performed using DH5-alpha competent *E. coli* cells and bacteria were grown at 37˚C.

Wild type D10 *P. falciparum* was transfected at the early ring stage with linearized template vector (pMG75, 50 μg) digested with EcoRV (NEB) overnight along with circular gRNA plasmids (40 μg each). Post transfection, the culture was maintained in media supplemented with 250 nM anhydrotetracycline (aTc, Millipore Sigma) for first two days, followed by blasticidin (BSD) + aTc media. The transfected parasite genotype was confirmed by PCR analysis. Details of cloning procedure is shown in Supporting information.

## Immunofluorescence assay (IFA)

A volume of 50 μL parasitized RBCs at trophozoite stage (~5% parasitemia) was collected for IFA. The parasites were labeled with 50 nM Red Mito-Tracker CMXRos (Thermo Fisher Sci-entific) for 20 mins. This was followed by three washes using 1X PBS before fixing the cells with 4% paraformaldehyde/ 0.0075% glutaraldehyde in PBS for 1hr at 37˚C on a rotator. The fixed cells were processed as described in our previous publication and visualized under Nikon Ti microscope [18]. The anti-HA antibody (Sc-7392, Santa Cruz) and secondary antibody (A11029, Invitrogen) were used in 1:300 dilution.

## Parasite growth assay

The endogenously tagged parasite lines were tightly synchronized with two rounds of 0.5 M alanine/10 mM HEPES (pH 7.4) at the ring stage. At the trophozoite stage, synchronized para-sites with high parasitemia (8–10%) were washed 3 times with regular RPMI to remove aTc. Washed cells were split 1:10 with new RBCs and were equally transferred to two T25 flasks to allow the parasite growth with and without aTc (250 nM). Parasite morphology and growth rates were monitored in 48 h intervals in Giemsa-stained thin blood smears. Parasitemia was counted per 1000 RBCs under Leica light microscope. Every 48h post KD, the culture was split 1:5 over several IDCs to collect protein samples for Western blot analysis. Growth curve data was analyzed using GraphPad Prism 9.

## Western blotting

Parasite culture was harvested at the trophozoite stage and treated with 0.1% saponin/PBS sup-plemented with 1X protease inhibitor (PI) (APEX Bio). The saponin treated parasites were washed with 1X PBS and PI to remove residual hemoglobin. The parasitized pellet was mixed thoroughly with 10 volumes of 3% SDS/60 mM Tris-HCl (pH 7.5) and solubilized overnight at 4˚C on a rotator. The sample was spun down at 23,000 g for 10 mins to acquire supernatant for SDS-PAGE. Protein concentration was determined by the Pierce BCA Protein Assay (23227, ThermoFisher). Post electrophoresis, protein was transferred to methanol activated PVDF membrane (0.45 μm, HybondTM) at 23V overnight at 4˚C. The membrane was blocked with 5% milk/PBS for 2 h before incubation with mouse monoclonal anti-HA antibody (Sc-7392, Santa Cruz) at 1:10,000 for overnight at 4˚C. Horseradish peroxidase (HRP) conjugated

secondary goat anti-mouse antibody (Cat: 62–6520, Thermo Fisher Scientific) at 1:10,000 was used for 4 h at room temperature. The membrane was incubated with Pierce ECL Western Blotting Substrate and visualized using the BioRad imager. The membrane was further probed with 1:10,000 rabbit anti-PfExp2 primary antibody followed by 1:10,000 HRP conjugated secondary anti-rabbit antibody (Thermo Fisher) for 1h, before it was developed to show loading controls.

## Hypoxanthine growth inhibition assay

Parasite lines were grown in the presence of 250 nM aTc and tightly synchronized by treatment with 0.5M alanine/10 mM HEPES (pH 7.4) at the ring stage. The synchronized trophozoites were used to initiate 48 h growth inhibition assays in 96-well plates for the PfRSM22 line on days 0, 2, 4 and for the PfMRPL23 line on days 0, 2. The schematic of the assay design is illustrated in Fig 2A. On day 0, the cultures were washed to remove aTc and subsequently grown in the presence or absence of aTc. On the days of assay setup (days 0, 2, 4), the cultures were administered with serially diluted antimalarials (atovaquone, ELQ-300, and chloroquine) for 48 h and [$^3$H] hypoxanthine for 24 h before being harvested. Other procedures of this assay have been described in our previous work [20]. The data represented was analyzed using GraphPad Prism 9.

## Mitochondrial enrichment from malaria parasites

PfRSM22 and PfMRPL23 KD parasite lines were tightly synchronized with 0.5 M alanine/10 mM HEPES (pH 7.4) at the ring stage and expanded into T175 flasks. KD was set up by removing aTc at the late trophozoite stage [18]. Mitochondrial enrichment was carried out by a modification of previously described methods, principally omitting the step of removing hemozoin by a magnetic column. Briefly, asexual stage trophozoites at 10% parasitemia were released from RBCs using 0.1% saponin/AIM buffer (120 mM KCl, 20 mM NaCl, 20 mM glucose, 6 mM HEPES, 6 mM MOPS, 1 mM MgCl2, 0.1 mM EGTA, pH 7.0). The harvested cells were washed with cold 1X AIM buffer to remove excess hemoglobin and other soluble components. The washed pellet was resuspended in ~30 mL mitochondrial isolation buffer (MSEH: 225 mM mannitol, 75 mM sucrose, 4.3 mM MgCl2, 0.25 mM EGTA, 10 mM HEPES [Tris], 5 mM HEPES [KOH], pH 7.4) and subjected to $N_2$ cavitation at 1000–1200 psi on ice followed by centrifugation at 900 x g to remove nuclei and unbroken debris. The supernatant was transferred to a prechilled tube on ice and centrifuged at 23000 x g for 30 min at 4˚C. The supernatant containing cytosol and low-density components was removed. The pellet containing enriched parasite mitochondria with other organelles was resuspended in a minimal volume of MSEH buffer enriched with 0.5 M ectoine and 0.5 M hydroxyectoine (biomolecule stabilizers) [21] and stored at -80˚C. Membrane protein complexes including the $bc_1$ complex were solubilized from the mitochondrial suspension using 0.7 mg n-Dodecyl β-D-Maltoside (Anatrace) per mg protein for 30 mins at 4˚C. The supernatant obtained from a 90 min spin at 19000 x g at 4˚C was used for $bc_1$ complex assays. Other debris including hemozoin in the pellet was discarded.

## Ubiquinol-cytochrome *c* oxidoreductase assay ($bc_1$ complex assay)

Mitochondria were enriched from trophozoites of the parasite lines grown in the presence or absence of aTc. The activity of complex III to reduce cytochrome *c* in the presence of ubiquinol was measured using CLARiTY, a light-scattering UV/VIS spectrometer (OLIS, Bogart, GA), according to our previously published work [22]. Briefly, in an assay volume of 320 μL (pH 7.4), 60 μM of oxidized horse heart cytochrome c (electron acceptor, Millipore Sigma) and 2

mM KCN (inhibiting complex IV activity) were incubated with n-Dodecyl β-D-Maltoside solubilized $bc_1$ complex at 35˚C, followed by addition of 100 μM decylubiquinol (electron donor; gift of Professor Michael Riscoe, Oregon Health and Science University) to initiate the reaction. The enzymatic reduction of cytochrome *c* over the next few minutes was recorded at 550 nm using the spectrophotometer.

## RNA extraction and library construction

D10-PfRSM22-3HA and D10-PfMRPL23-3HA parasite lines were tightly synchronized and grown in T175 flasks in presence of aTc. At the trophozoite stage (day 0), the parasites were washed and split into new RBCs to start KD. The parasites were harvested at late trophozoite stage on day 2, day 4 and day 6 by saponin lysis (0.05% in PBS). The saponin lysed parasite pellets were immediately dissolved in 6 volumes of TriZol (Invitrogen) and kept frozen at—80˚C until to be thawed on ice for RNA isolation. For each sample, 1/5$^{th}$ volume of chilled chloroform was added, followed by a spin in 4˚C at 2000 g for 10 mins. The top aqueous layer was transferred to a fresh chilled tube without disturbing the middle buffy coat. 100% ethanol was added to the aqueous layer in a 1:1 ratio. This mixture was subjected to RNA isolation using a kit according to the manufacture's protocols (Zymo research). The concentration of eluted RNA was determined using Nanodrop (Thermo Fisher). Quality of eluted RNA was tested by running 1μg of RNA on agarose gel at 4˚C to visualize intact cytosolic rRNA. Intact rRNA bands were confirmed indicating no obvious RNA degradation.

Isolated RNA samples were sent to Novogene Co for sequencing where the samples were further quantified using a qubit fluorometer (Thermo Fisher, Waltham, MA) and the quality was scored using a bioanalyzer (Agilent Technologies, Santa Clara, CA). Samples with RIN 8.0 and above were selected for library preparation. Briefly, 1μg of input RNA was used for RNA library preparation using the NEB Next Ultra II RNA Library Prep kit for Illumina (E7775L; NEB). Prepared libraries were quantified using real time PCR and bioanalyzer. Libraries of 3 nM and above concentration were loaded on to NovaSeq 6000 S4 Reagent Kit using a paired-end 150 kit (20012866; Illumina, San Diego, CA). Each sample was sequenced to a depth of at least 19.5 million reads. The number of clean reads obtained from each sample is listed in S5 Table.

## RNA sequencing and data analysis

The fastq files were generated using HWI-ST1276 instrument. Quality of reads was checked using FastQC tool and later processed to remove the adapter sequences and to perform base quality control check [23]. Quality metrics and error rate were transformed using Phred score with highest stringencies [24]. Fastp processed files were aligned to the most updated *P. falciparum* reference genome using HISAT2 software (v2.0.5) [25]. Exons comprised of more than 90% of the sequences. RPKM counts of each gene was counted using Stringtie (v1.3.3) [26]. Differential expression between control and KD samples was calculated using DEseq2 R package (v1.20.0) [27]. Samples were processed as duplicates and analyzed individually for differential expression. The adjusted p value was calculated after correction of p value using Benjamini and Hochberg method. A p-value < 0.05 was considered to obtain significant DESeq2 values. KEGG (Kyoto Encyclopedia of Genes and Genomes (http://www.kegg.jp/)) pathway analysis was performed using clusterProfiler R package (v3.8.1) [28]. GO enrichment analysis was preformed using VEuPathDB tools in PlasmoDB [29]. GSEA analysis was performed using gsea v3.0. Volcano plots, heat maps and other graphs were made on GraphPad Prism 9.

## RT-qPCR

5 µg of total RNA was used to obtain cDNA using oligo-dT primers and superscript IV RT kit (Thermo Fisher). The cDNA was used for quantitative PCR using SYBR green master mix (Thermo Fisher) and gene specific primers under standard qPCR conditions (Applied Biosystems). Ct values obtained from each gene at each condition (+ or–aTc) were normalized to respective GAPDH controls. The fold change of mt rRNA abundance in the aTc minus compared to aTc plus cultures was then calculated according to $2^{-\Delta\Delta Ct}$ method. The data obtained was graphed using GraphPad Prism 9.

## Results and discussion

### PfRSM22 and PfMRPL23 are essential mitochondrial proteins

PfRSM22 (Pf3D7_1027200) is currently annotated as a putative MRP S22 in *Plasmodium* database (www.PlasmoDB.org), but contains an S-adenosyl-L-methionine (SAM) dependent methyltransferase like domain (InterPro IPR029063) [30]. Its ortholog in *Homo sapiens* is annotated as methyltransferase like protein 17 (METTL17); however, orthologues of PfRSM22 found in yeast, *Tetrahymena*, *Trypanosoma*, and *Toxoplasma* are annotated as RSM22 [31–33]. The methyltransferase activity of this protein is known to play a role in maturation of mt rRNA and mitoribosomal assembly [30]. *Trypanosoma brucei* RSM22 interacts with immature mt rRNA and is a member of the early SSU assemblosome. Its KD leads to reduction of the 9S rRNA (SSU mt rRNA) expression level, although its association with the mature mitoribosome has not been observed by cryo-EM [33]. We propose that Pf3D7_1027200, although annotated as putative MRPS22 in PlasmoDB, should be termed RSM22, a putative mitochondrial ribosome assembly protein on the basis of its resemblance to the InterPro Ribosomal protein RSM22-like protein family (IPR015324/PF09243) with strong statistical support (E = 0.0) (S1 Fig) [34]. On the other hand, mS22, found in metazoan mitoribosomes, belongs to a different protein family that lacks methyltransferase activity (InterPro IPR019374).

PfMRPL23 (Pf3D7_1239100) likely belongs to LSU based on its homology to L23 proteins, which are highly conserved throughout all kingdoms of life (InterPro Ribosomal protein uL23 family IPR013025/Pfam PF00276, E = 1.1e-13 for the PfMRPL23 match) (S2 Fig). Previous reports represent that *E. coli* L23 interacts with domain III of the LSU rRNA and is a part of the exit tunnel of the ribosome [35, 36]. The exit tunnel is a crucial site for the mitoribosome as it is involved in co-translational insertion of highly hydrophobic mtETC proteins into the mitochondrial inner membrane [37].

To investigate the essentiality of PfRSM22 and PfMRPL23, we genetically modified their loci in the *P. falciparum* D10 WT parasite line using a CRISPR/Cas9 mediated double crossover recombination strategy. PfRSM22 and PfMRPL23 loci were individually modified by integrating 3xHA epitope tags and 8 copies of TetR binding RNA aptamers at the 3' end of the coding sequence, providing controlled expression of the gene using the TetR-DOZI-aptamer system [38]. Positively transfected parasites were obtained in 4 weeks post transfection under constant pressure of blasticidin and anhydrotetracycline (aTc, 250 nM). Genotyping of the transgenic lines, D10-PfRSM22-3HA and D10-PfMRPL23-3HA, was performed using specific primers (S1 Table). The strategy of integration, confirmation of modified loci and the absence of the wildtype gene are shown in S3 Fig. Intracellular localization of PfRSM22-3HA and PfMRPL23-3HA was examined by immunofluorescence assay (IFA). Both proteins co-localized with the fluorescent mitochondrial probe Mitotracker red (Fig 1A and 1B), indicating that PfRSM22 and PfMRPL23 are mitochondrial proteins.

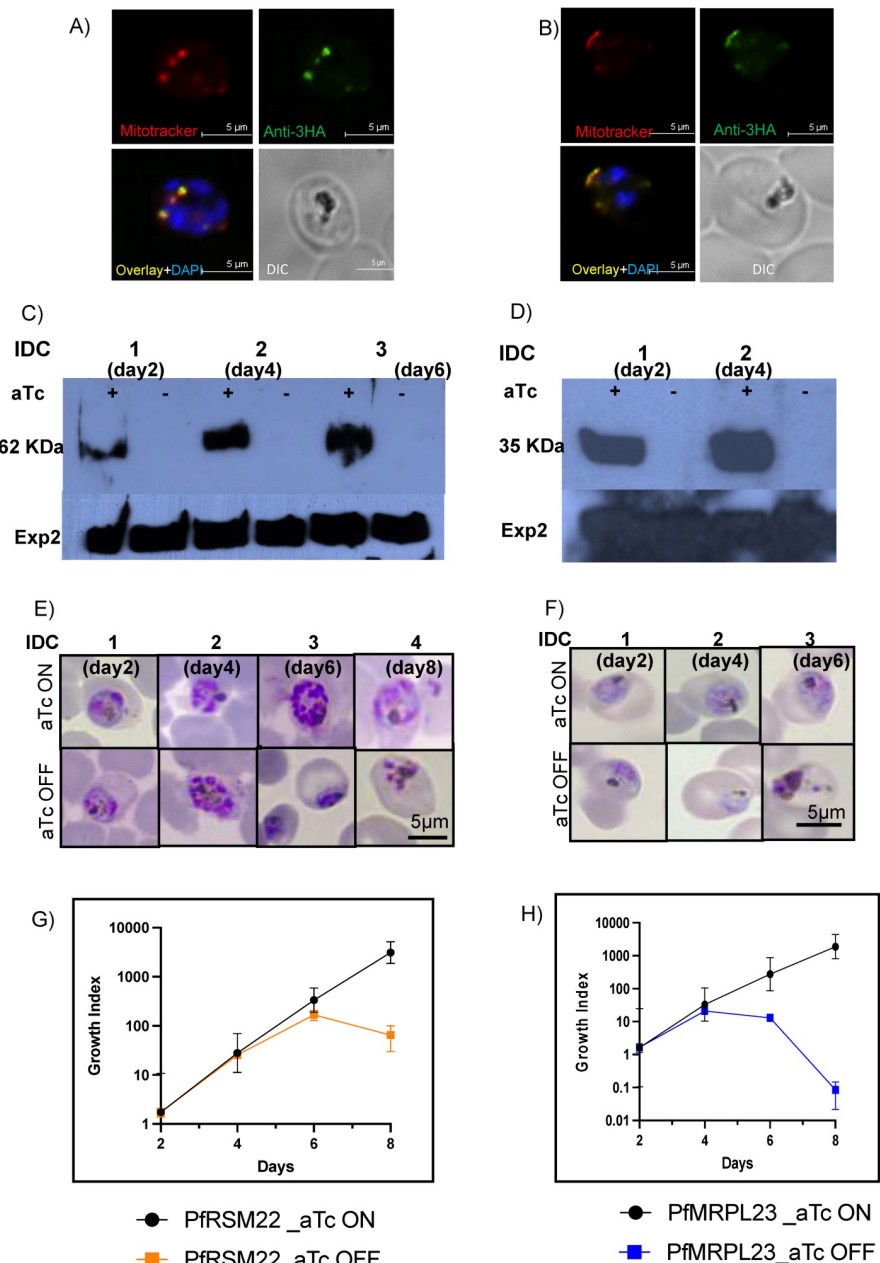

**Fig 1. PfRSM22 and PfMRPL23 are localized to the mitochondrion and are essential for parasite survival.** (A) Images from fluorescence microscopy showing colocalization of PfRSM22-3HA (green) with Mitotracker (red). (B) Colocalization of PfMRPL23-3HA (green) with Mitotracker (red). In A and B, the parasite nucleus is stained using DAPI (blue). (C) Western blot showing expression of PfRSM22-3HA protein (62 KDa) in the presence or absence of 250 nM aTc over 6 days (3 IDCs). (D) Western blot showing expression of PfMRPL23-3HA protein (32 KDa) in the presence or absence of 250 nM aTc over 4 days (2 IDCs). Pf-Exp2 (33 KDa) was used as a loading control. Representative images of Giemsa-stained D10-PfRSM22-3HA parasites (E) and D10-PfMRPL23-3HA parasites (F) at the trophozoite stage in the presence (aTc ON) and absence of aTc (aTc OFF). Quantification of D10-PfRSM22-3HA parasite growth (G) and D10-PfMRPL23-3HA parasite growth (H) over 8 days in the presence and absence of aTc. Growth index was calculated by multiplication of parasitemia with splitting factors over the time course. Data shown here is mean ± SD of n = 3.

Upon removal of aTc from the culture medium to induce KD of PfRSM22 or PfMRPL23, we assessed parasite viability and morphology over 8 days or 4 intraerythrocytic cycles (IDCs).

The parasites were grown in medium containing 250 nM aTc to allow expression of the respective proteins (aTc ON). As shown in Fig 1C and 1D, PfRSM22 and PfMRPL23 were expressed in the presence of aTc, and within 2 days of aTc removal, neither proteins could be detected by Western blotting. The effects on parasite morphology observed over time following KD of PfRSM22 and PfMRPL23 are shown in Fig 1E and 1F using light microscopy of Giemsa-stained thin blood smears. Late-stage parasites grown without aTc were morphologically abnormal compared to aTc plus parasites by day 6 in the D10-PfRSM22-3HA line and by day 4 in the D10-PfMRPL23-3HA line. It was interesting to note the difference in progression to death upon KD of PfRSM22 vs PfMRPL23. We found that D10-PfRSM22-3HA survived for 6 days and appeared dead on the 8th day after aTc removal, while D10-PfMRPL23-3HA parasites died on day 6 following aTc removal. Overall, we show that both PfRSM22 and PfMRPL23 are essential for the survival and development of asexual stage parasites. However, PfRSM22 KD parasites were able to survive for one IDC more than the PfMRPL23 KD parasites. Quantification of parasitemia in both KD lines is shown in Fig 1G and 1H.

Based on these results, we have categorized the effects of knocking down PfRSM22 or PfMRPL23 into two categories, the early phase (day 2 aTc off) and the late phase (day 6 aTc off for PfRSM22 vs day 4 aTc off for PfMRPL23). In both parasite lines, morphological and growth defects became evident only in the late phase after KD, which is consistent with KD studies of other PfMRPs [17, 18]. In the early phase after KD, however, both KD parasite lines appeared healthy despite PfRSM22 and PfMRPL23 proteins being undetected by Western blotting. We reasoned that the effects of PfRSM22 or PfMRPL23 KD were probably not lethal until the preexisting protein translation products (mtETC proteins) have been entirely lost to protein turnover. Previous studies have shown that mitochondrial respiration occurs at a low level in asexual blood stages [39] and functions mainly to serve pyrimidine biosynthesis by recycling reduced ubiquinone [40]. Thus, continued inheritance of previously assembled mtETC complexes can serve this function for at least one IDC without the need for newly assembled mitoribosomes.

## PfRSM22 and PfMRPL23 KD leads to mitochondrial functional defects

We reasoned that KD of PfRSM22 or PfMRPL23 would affect mitochondrial protein translation and lead to reduced production of mitochondrially translated proteins. However, due to the lack of antibodies against any of the three proteins, *cyt b*, COXI, and COXIII, direct measurements of mitochondrial protein translation remain difficult at present. To overcome this challenge, we assessed mitochondrial functions in the knockdown parasites with two surrogate methods. The first one measures parasite growth under the combination of genetic KD and chemical inhibition that induce synthetic lethality. If KD of PfRSM22 or PfMRPL23 led to defects in mitochondrial protein translation, the KD parasites would become hypersensitive to $bc_1$ inhibitors since *cyt b* is translated by the mitoribosome. The second method directly measures the $bc_1$ enzymatic activity from isolated mitochondria of KD parasite lines using a spectrometer.

We performed [3H] hypoxanthine growth inhibition assays in both parasite lines using synchronized cultures at the trophozoite stage. On day 0, the parasite cultures were thoroughly washed to remove aTc. They were then exposed to serially diluted antimalarials, atovaquone, ELQ-300, or chloroquine, in the presence or absence of aTc (250 nM). PfRSM22 and PfMRPL23 parasite cultures were grown for 3 or 2 IDCs as indicated (Fig 2A). At each time point, radioactive [3H] hypoxanthine was administered to the cultures 24 hours before harvest (Fig 2A). In the PfRSM22 KD line, we observed a reduction in IC50s of atovaquone and ELQ-300 from day 2 to day 6 in comparison to the aTc plus cultures (Fig 2B and 2D). A similar

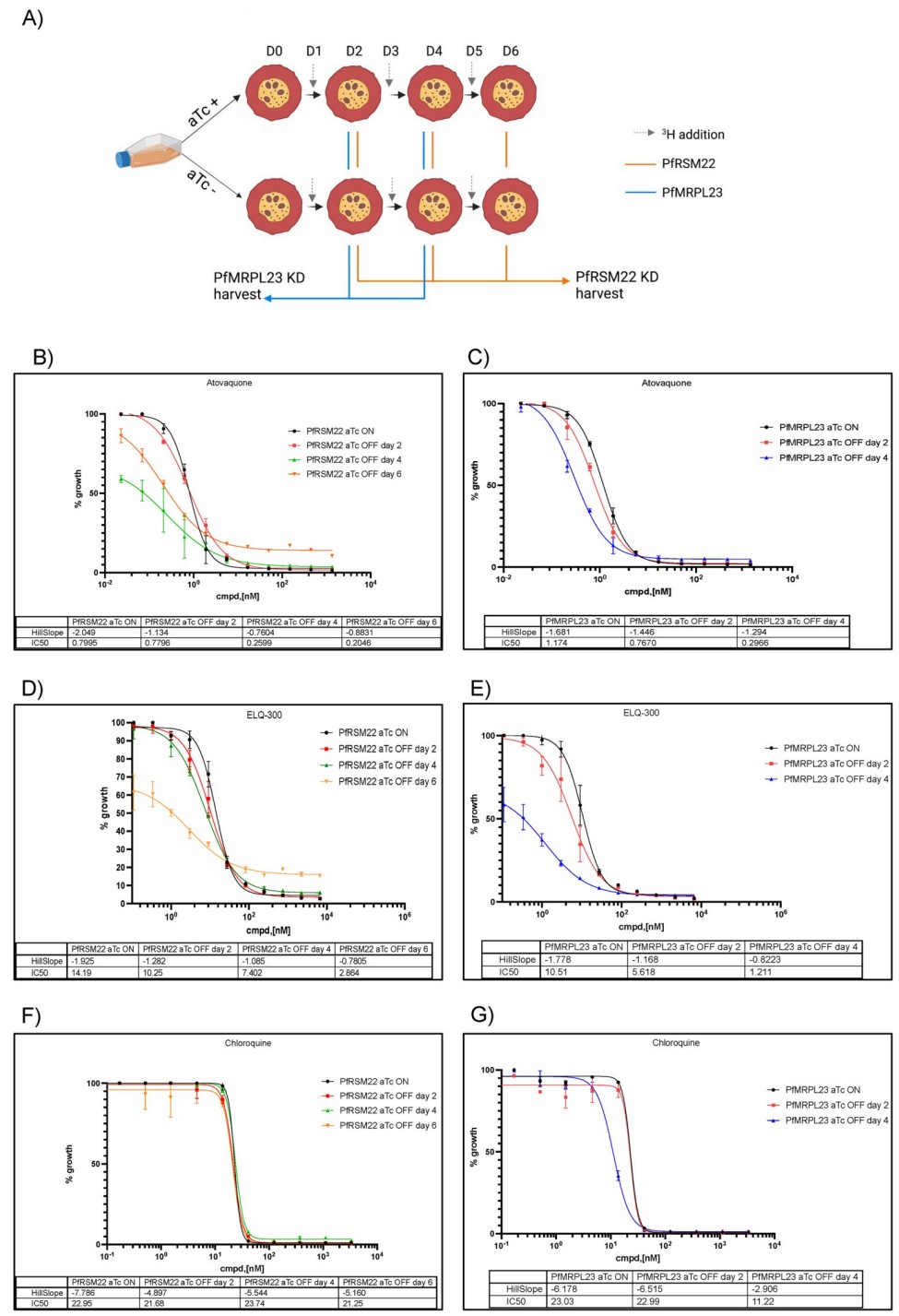

**Fig 2. PfRSM22 and PfMRPL23 KD increased sensitivity to antimalarials targeting the parasite *bc1* complex.** (A) The schematic represents key steps of growth inhibition assays measured by [³H] hypoxanthine incorporation. The diagram was created using BioRender.com. (B and C) Atovaquone hypersensitivity upon PfRSM22 and PfMRPL23 KD respectively. (D and E) ELQ-300 hypersensitivity upon PfRSM22 and PfMRPL23 KD respectively. Reduction in IC50 values and Hill Slope of the curve were reported for respective parasite lines. Data shown are the mean ± S.D. of triplicates from n = 3 independent experiments. (F and G) KD of PfRSM22 or PfMRPL23 did not show hypersensitivity to chloroquine.

trend of hypersensitivity to these two antimalarials was also observed in the PfMRPL23 line on day 2 and day 4 post KD (Fig 2C and 2E). In contrast, PfRSM22 and PfMRPL23 parasite lines did not exhibit hypersensitivity to chloroquine, an antimalarial targeting hemoglobin digestion, in early or late days post KD, although PfRMPL23 KD on day 4 showed a slightly increased sensitivity to chloroquine (Fig 2F and 2G). Overall, these data have shown that KD of PfRSM22 or PfMRPL23 leads to hypersensitivity to antimalarials targeting the $bc_1$ complex, suggesting that PfRSM22 are PfMRPL23 are involved in mitochondrial protein translation.

Next, we measured the $bc_1$ complex enzymatic activity using a spectrophotometric assay previously established in our laboratory with modifications (Materials and methods). This assay directly measures the reduction of cytochrome *c* by the $bc_1$ complex in the crudely purified mitochondrial samples (Fig 3A), serving as an alternative to assess mitochondrial protein translation due to the lack of antibodies against Pf mtDNA encoded proteins [41, 42]. We observed a decrease in the $bc_1$ enzymatic activity in the mitochondrial samples obtained from PfRSM22 and PfMRP23 KD lines (Fig 3B and 3C). This result is in accord with our previous publications showing reduced $bc_1$ activity upon disruption of other mitoribosomal proteins [17, 18]. This reflects reduced efficiency of mitochondrial protein translation resulting in reduced mtETC activity. Our current and previous results consistently show that although *P. falciparum* can sustain reduction of $bc_1$ complex activity up to 50%, further decline in its efficacy limits parasite progression to next cycle, leading to parasite demise (Fig 3B and 3C). PfMRPL23 KD had a more dramatic effect on complex III activity, only allowing the parasite to survive for two IDCs in the absence of newly synthesized PfMRPL23 protein. On the other hand, PfRSM22 KD did not cause more than 50% decline in $bc_1$ activity until day 6 after aTc removal, allowing this parasite line to continue to survive for one more cycle compared to PfMRPL23 KD. This difference could indicate that PfRSM22 and PfMRPL23 play different roles in mitochondrial protein translation. As discussed above, RSM22 is likely an early assembly factor, while L23 is a critical component of the exit tunnel of ribosomes. Theses speculations deserve further experimental characterizations in the future.

## Transcriptomic changes upon KD of PfRSM22 or PfMRPL23 in the early phase

To investigate global transcriptional changes upon loss of essential mitoribosomal proteins, we tightly synchronized both the D10-PfRSM22-3HA and D10-PfMRPL23-3HA lines and initiated KD by removing aTc from trophozoite stage parasites (day 0). We performed RNA sequencing of total poly A+ RNA isolated from PfRSM22 KD parasites on days 2, 4 and 6, and PfMRPL23 KD parasites on days 2 and 4 (Materials and methods). RNA samples isolated from D10-PfRSM22-3HA and D10-PfMRPL23-3HA parasite lines grown in the presence of aTc for two days were used as controls for RNA sequencing. Normalized read counts and DEseq2 values of the RNA sequencing results are provided in S2 Table. Differentially regulated genes with statistical significance (Benjamini-Hochberg p-adjusted value) below 0.05 were selected to generate Volcano plots.

Volcano plots represent statistical significance (p adj) vs the magnitude of changes (log2 fold change) of all gene transcripts in PfRSM22 aTc OFF vs PfRSM22 aTc ON samples and PfMRPL23 aTc OFF vs PfMRPL23 aTc ON samples. Each dot represents one gene in the Volcano plots. On day 2 post aTc removal, we observed significant transcriptomic alterations in both KD parasite lines. Fig 4A depicts that within 2 days of PfRSM22 KD, 136 genes were upregulated by $\geq$ 2-fold and 581 genes were downregulated by $\geq$ 2-fold (black dots). On the other hand, within 2 days of PfMRPL23 KD, 728 genes were upregulated, and 989 were downregulated by $\geq$ 2-fold (Fig 4B). Next, we performed KEGG pathway analysis of significantly

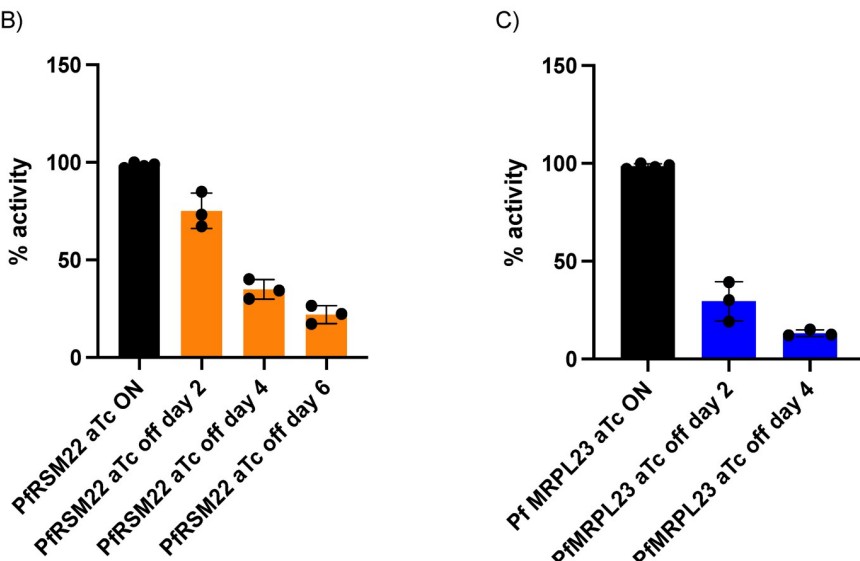

**Fig 3. Reduced cytochrome $bc_1$ complex activity upon PfRSM22 and PfMRPL23 KD.** (A) Schematic presentation of the *Plasmodium* mtETC. Complex III ($bc_1$ complex) activity was measured while complex II and complex IV were inhibited using malonate and KCN respectively. Modified from our previous publication [10]. Created with BioRender.com. Cyt *c* reduction was measured at 550 nm representing functionality of $bc_1$ complex in enriched and solubilized mitochondria from PfRSM22 KD (B) and PfMRPL23 KD (C). The $bc_1$ enzymatic activity of the knockdown parasites in PfRSM22 (orange) and PfMRPL23 (blue) was normalized to their respective aTc ON controls. Kruskal-Wallis test in n = 3 experiments. ***, $p < 0.0001$; **, $p < 0.001$.

regulated genes (p adj< 0.05) upon KD of PfRSM22 or PfMRPL23 on day 2. In PfRSM22 KD, we observed downregulation of 23 genes that belong to the broader KEGG pathway representing proteasome catabolic processes, and 80 genes that belong to the KEGG pathway representing ribosomes, RNA binding proteins and translation related proteins (Fig 4C, orange). The PfMRPL23 KD downregulated a wider range of KEGG pathways on day 2 including pentose phosphate pathway, RNA degradation and spliceosome proteins, along with the ribosomal biogenesis pathway (Fig 4C, blue). Among the upregulated pathways on day 2, on the other

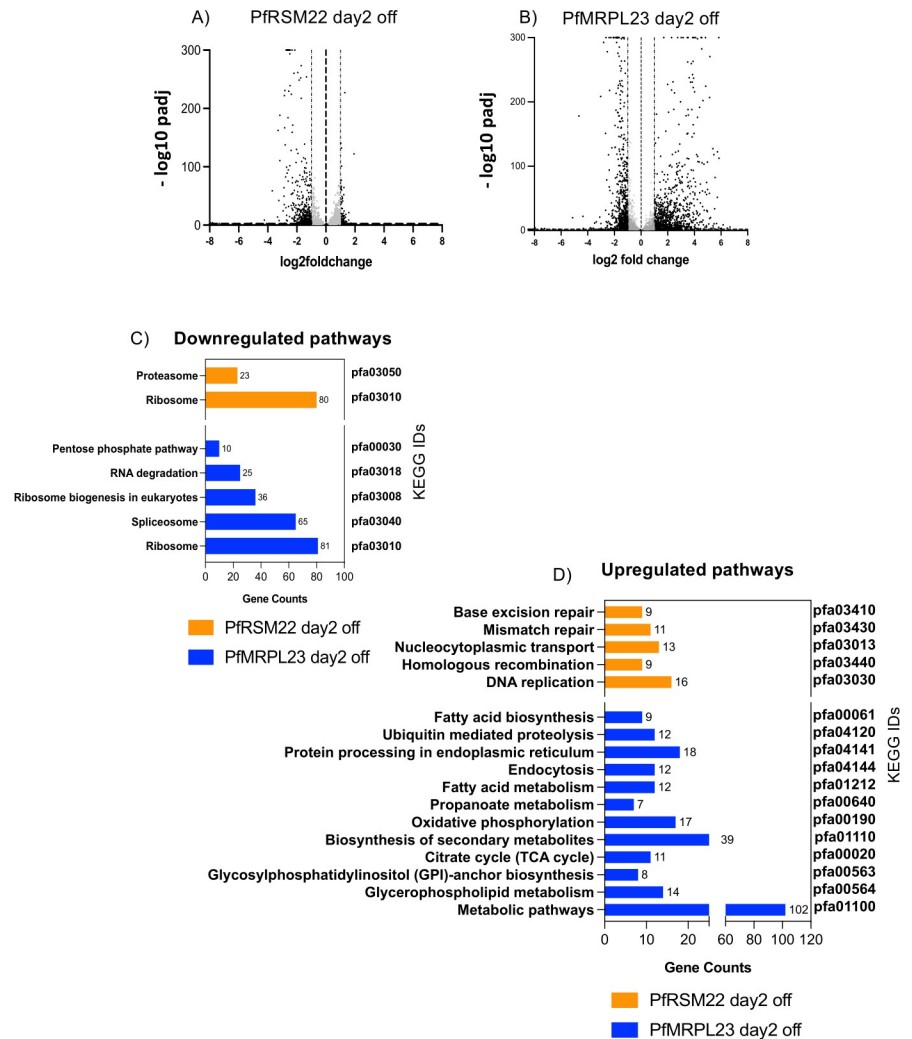

**Fig 4. Transcriptomic changes in the early phase after KD of PfRSM22 and PfMRPL23.** Volcano plots of differentially regulated transcripts (p adj <0.05) on day 2 post KD in the PfRSM22 KD line (A) and in the PfMRPL23 line (B). Each black dot in the Volcano plots represents a gene regulated at least by 2-fold (log2 fold $\geq$ 1 and $\leq$ -1). Each grey dot indicates a gene differentially regulated less than 2-fold (log2 fold <1 and > -1). (C) Significantly downregulated KEGG functional pathways based on the number of genes (gene counts) following the KD of PfRSM22 or PfMRPL23 on day 2. (D) Significantly upregulated KEGG functional pathways based on the number of genes (gene counts) following the KD of PfRSM22 or PfMRPL23 on day 2.

hand, KD of PfRSM22 resulted in a slight upregulation of transcripts involved in DNA metabolic processes and nucleocytoplasmic transport (Fig 4D, orange). KD of PfMRPL23 showed an increase in multiple metabolic processes, such as fatty acid metabolism, TCA cycle, protein processing in ER, oxidative phosphorylation, glycerophospholipid metabolism, biosynthesis of secondary metabolites and other metabolic pathways (Fig 4D, blue). The gene IDs of down-or-up regulated KEGG pathways are shown in S3 Table. In addition, we also performed GO enrichment analysis using the tools available in PlasmoDB (www.PlasmoDB.org) and the differentially regulated biological pathways are represented in S3 Table.

Overall, these data suggest that KD of PfRSM22 and PfMRPL23 leads to dramatic changes at the transcriptional level as early as day 2. Despite these transcriptional changes, parasites appeared healthy without any noticeable morphological defects (Fig 1E and 1F).

## Early effects of PfRSM22 and PfMRPL23 KD on transcripts of mitoribosomal components

There are 43 annotated *Plasmodium* MRPs, of which 14 are SSU proteins and 29 are LSU proteins, based on homology with bacterial ribosomes and mitoribosomes of other eukaryotes. This is close to half the number of proteins known in yeast and mammalian mitoribosomes [43]. The MRPs have been categorized into early, intermediate and late assembly proteins based on their hierarchical association at different time points forming multiple pre-ribosomal complexes [43, 44]. So far, nothing is known about the mitoribosomal assembly processes in *P. falciparum* due to technical limitations associated with isolating mitochondria or mitoribosomes in large quantities. However, previous research in other organisms describe the details of mitoribosomal assembly processes [43–45]. RSM22 is known to play a role in early mitoribosomal assembly and maturation of mt SSU [33]. L23 is one of the intermediate binding proteins that makes few or no contact with early assembly proteins, but their incorporation into the ribosome depends on stability of early assembly proteins [46].

We found that 24 out of 43 annotated PfMRPs were significantly downregulated on day 2 KD of PfRSM22; among these, 10 belonged to the SSU and 14 belonged to the LSU (Fig 5A, orange). No PfMRP transcript was upregulated upon PfRSM22 KD. PfMRPL23 KD led to downregulation of transcripts encoding 19 PfMRPs (6 of SSU and 13 of LSU) and upregulation of 5 PfMRPs, of which 2 belonged to the SSU (S8 and S14) and 3 were LSU PfMRPs (L11, L16 and L19) (Fig 5A, blue). These data suggest that, in the early phase, PfRSM22 KD resulted in downregulation of most PfMRPs, whereas PfMRPL23 KD caused upregulation of some PfMRPs.

A high sequencing depth of our RNAseq analysis and presence of a poly A tail in most mt rRNA fragments facilitated investigation of these transcripts in this study. The PfRSM22 KD resulted in decreased levels of 11 SSU fragments, 15 LSU fragments, and 8 RNA fragments with unknown identity (Fig 5B, orange). The remaining 5 mt rRNA fragments (RNA16, RNA20, SSUA, RNA27t, LSUB) could not be detected in our RNA sequencing results. The positions of all mt rRNAs on the ribosomal secondary structure proposed in previous studies [11] are shown in S5 Fig. Previous studies in mammals and *Trypanosoma* have shown that disruption of RSM22 only affected expression of SSU mt rRNA, but not LSU mt rRNA [33, 46]. Upon KD of PfRSM22, however, expression of both SSU and LSU mt rRNA transcripts was downregulated. This could be due to the scrambled arrangement of these mt rRNA genes in the *P. falciparum* 6 kb mtDNA. In contrast to PfRSM22 KD, PfMRPL23 KD on day 2 led to upregulation of 8 SSU mt rRNA transcripts, 10 LSU mt rRNA transcripts, and 6 RNA transcripts of unknown identity (Fig 5B, blue). No mt rRNA genes were significantly downregulated upon KD of PfMRPL23 on day 2. Perhaps the mtDNA transcription was stabilized or upregulated upon loss of PfMRPL23, but not PfRSM22. This could be due to their hierarchical roles in the formation of mitoribosomal complexes at different time points. While PfRSM22 is expected to be an early mitoribosomal assembly factor, PfMRPL23 might be involved in the mitoribosomal assembly process at later time points as suggested in studies of other mitoribosomes [45]. To further verify the transcriptional changes of mt rRNA fragments in the KD lines, we performed RT-qPCR in representative mt rRNA genes that are relatively large in sizes (80–200 bases). The differential expression of 2 SSU (RNA5, 8) and 2 LSU (LSUA, RNA10) mt rRNA genes in the KD lines is shown in S4 Fig. RT-qPCR revealed that these mt rRNA genes were regulated in a similar trend as shown in RNAseq analysis,

In addition, our RNA sequencing data further confirm some intriguing aspects of mtDNA transcription in malaria parasites that were noted previously [12]. Individual mt rRNA fragments do not appear to be expressed in stoichiometric levels, suggesting differential expression

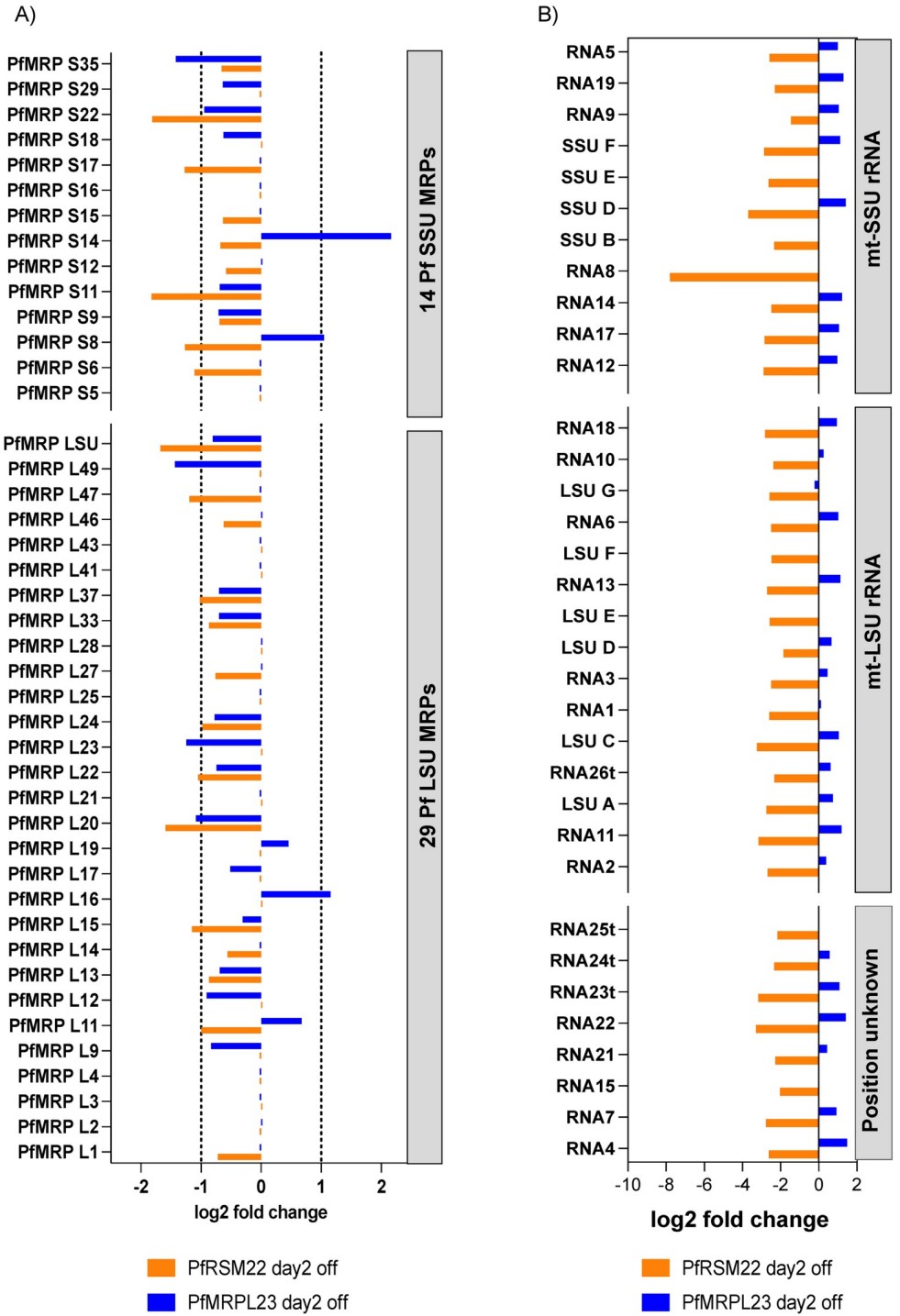

**Fig 5. Early effects of PfRSM22 and PfMRPL23 KD on transcripts of mitoribosomal components.** (A) Log2 fold change of putative PfMRPs on day 2 KD of PfRSM22 (orange) or PfMRPL23 (blue). (B) Log2 fold change of mt rRNA fragments on day 2 KD of PfRSM22 (orange) or PfMRPL23 (blue). The positions of mt rRNA transcripts are shown in S5 Fig.

profiles of these transcripts as well as their contributions to mitoribosomal assembly in *Plasmodium*. Furthermore, our study detected 34 out of 39 fragmented rRNA molecules being

poly A positive, which was largely in agreement with the previous results of Feagin and colleagues [13] showing the presence of post transcriptional polyadenylation of mt rRNA transcripts. Unlike transcripts of the apicoplast genome, mt rRNA genes are not only fragmented but also post-transcriptionally polyadenylated. However, the exact function and enzymes participating in the process of polyadenylation of mt rRNAs remain unknown.

Moreover, we checked the transcriptional changes in the genes related to the parasite apicoplast. We found that four nuclearly encoded apicoplast ribosomal proteins (PfARPs; S14, L12, L18 and L35) and several transcripts encoding proteins targeted to the apicoplast were also downregulated in the early phase of PfRSM22 and PfMRPL23 KD (S6 Fig panel A). Some PfARPs are, however, encoded in the apicoplast genome, and due to the absence of polyadenylation in apicoplast transcripts, the effect of PfRSM22 and PfMRPL23 KD on apicoplast encoded transcripts remained unexamined here.

Together, in the early phase of PfRSM22 and PfMRPL23 KD, transcripts of mitoribosomal components were significantly perturbed. However, the findings from this study are limited due to the current incomplete picture of MRPs in *Plasmodium*. Our results do not account for unknown MRPs that undoubtedly remain undiscovered in apicomplexan parasites. In addition, PfMRPL23, like a number of other PfMRPs, contains additional amino acid sequences that are unique to apicomplexan parasites (S2 Fig) [34]. The role of parasite specific regions in these proteins, and their interactions with other proteins as well as fragmented mt rRNA remain unexplored at this point.

## Transcriptomic changes upon PfRSM22 and PfMRPL23 KD in the late phase

To understand the transcriptomic changes in the KD parasites right before the onset of parasite death, we carried out RNA sequencing of the PfRSM22 line on day 6 and PfMRPL23 line on day 4 post aTc removal. We chose to compare these late phase time points after KD of the respective proteins based on similarities in parasite growth decline and morphology. At these time points (day 6 of PfRSM22 KD and day 4 of PfMRPL23 KD), the parasites appeared healthy (Fig 1E and 1F). The transcriptome of PfRSM22 on day 4 after KD was also examined (included in S4 Table) but it is not included in the comparison here since there were only a few transcripts that altered compared to data of the day 2 after KD.

The overall profile of transcripts upregulated and downregulated upon PfRSM22 KD on day 6 and PfMRPL23 KD day 4 are shown in the Volcano plots in Fig 6A and 6B. Two downregulated KEGG pathways in common to both KDs at the later time points were related to the proteosome and oxidative phosphorylation (OxPhos) (Fig 6C and S3 Table). Downregulation of proteosome related functions suggests defective protein homeostasis at the late phase of KD. Reduction of transcripts related to OxPhos pathway was likely a downstream effect of collapsing mtETC components due to KD of PfRSM22 and PfMRPL23. These data indicated a series of events emanating from dysregulation of the mitochondrion in both parasite lines. These observations are aligned with our previous studies that showed severe defects in cytochrome $bc_1$ complex activity coinciding with noticeable morphological changes upon KD of other essential PfMRPs [17, 18]. Interestingly, PfMRPL23 KD on day 4 also caused downregulation of transcripts involved in metabolic pathways, biosynthesis of secondary metabolites, fatty acid metabolism, GPI anchor biosynthesis, protein processing in ER and biosynthesis of amino acids (Fig 6C, Blue). These findings are consistent with parasites being on the cusp of demise as their metabolism is throttled in response to mitochondrial disruption. Fig 6D represents the KEGG pathways upregulated after PfRSM22 KD on day 6 and after PfMRPL23 KD on day 4, which include transcripts related to cytoplasmic ribosome biogenesis. Specific upregulated

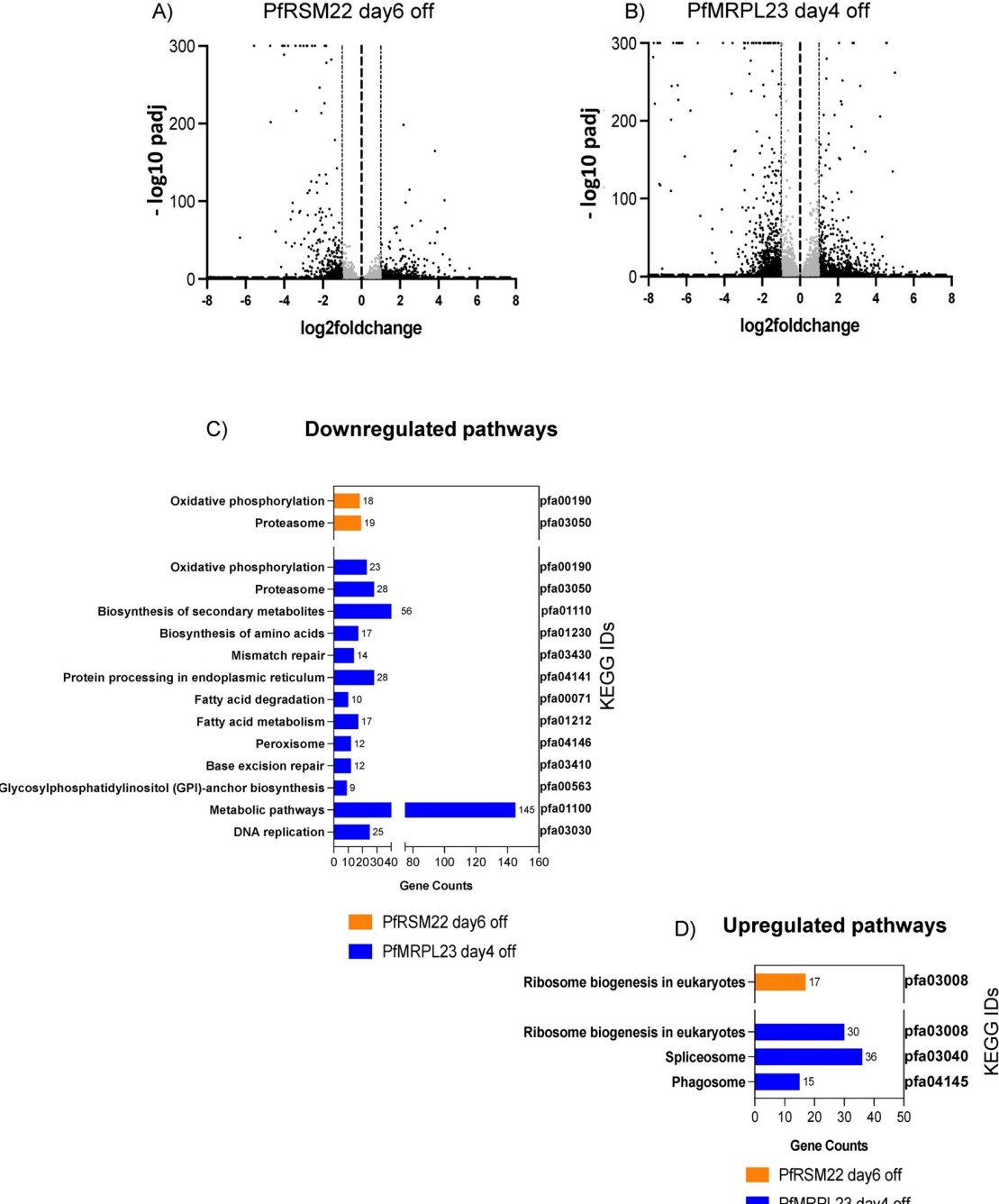

**Fig 6. Transcriptomic changes in the late phase KD of PfRSM22 and PfMRPL23.** Volcano plots of differentially expressed transcripts in the PfRSM22 KD parasite line on day 6 (A) and in the PfMRPL23 KD parasite line on day 4 (B) post aTc removal. Each black dot in the volcano plots represents a gene differentially regulated (log2 fold change ≥ 1 and ≤ -1). Each grey dot represents a gene differentially regulated with log2 fold change <1 and >-1. (C) KEGG functional pathways based on the number of genes (gene counts) that belong to a pathway downregulated due to PfRSM22 KD on day 6 and PfMRPL23 KD on day 4. (D) KEGG functional pathways upregulated based on the number of genes (gene counts) on day 6 of PfRSM22 KD and on day 4 of PfMRPL23 KD.

KEGG pathways due to PfMRPL23 KD on day 4 included transcripts related to spliceosome and phagosome. In addition, the gene IDs of down-or-up regulated KEGG pathways at the late

phase KD of PfRSM22 and PfMRPL23 are shown in S3 Table. We also performed GO enrichment analysis using the tools available in PlasmoDB (www.PlasmoDB.org) and the differentially regulated biological pathways are represented in S3 Table.

We further looked into the effects of the late phase KD on transcripts related to the parasite mitochondrion with the help of current information on the *P. falciparum* mitochondrial proteome. Although the complete mito-proteome of *P. falciparum* remains to be established, recent investigations of organellar proteins based on fractionation and bioinformatic approaches have generated lists of potential nuclearly-encoded proteins that are likely translocated to the parasite mitochondrion. We selected 475 proteins as a putative (unavoidably incomplete) mito-proteome based on manual annotation [47], *Plasmodium* orthologues of *Toxoplasma* mito-proteome [48, 49] and selected proteins listed in PlasmoMitoCarta [16]. This curated list along with 42 mtDNA encoded transcripts is denoted "mitochondria related transcripts" herein. This list of 517 genes in total (S4 Table) were used to assess the mitochondrial related transcripts significantly altered in the late phase following KD of each parasite line.

Fig 7A shows Venn diagrams of significantly downregulated transcripts (log2 fold ≤ -1) observed on day 6 following PfRSM22 KD, on day 4 following PfMRPL23 KD, and in the mitochondria (S4 Table). There were 186 common genes downregulated more than 2-fold on day 6 of the PfRSM22 KD and day 4 of the PfMRPL23 KD. Sixty-eight of those 186 commonly downregulated transcripts are related to mitochondria, including mt rRNA transcripts, the peptide release factor PfICT1 [50], translation initiation factor IF-3, PfMRPL20, PfMRPL25, etc. Fig 7B is a heat map representing log2 fold change (log2 fold ≤ -1) in the expression levels of these 68 mitochondrially related transcripts. Together, this list suggests defects in mitochondrial protein translation, leading to the downregulation of all three mtDNA encoded mRNA transcripts. The other mtETC and OxPhos transcripts listed in the heatmap are encoded in nuclear genome which together suggests a gradual collapse of transcripts essential for the parasite mtETC. Downregulation of these transcripts appears to be a common consequence of genetic ablation of PfMRPs in *P. falciparum*. Several proteins with unassigned functions also had reduced transcript levels following late phase KD. Analysis of these proteins and their tentative functional assignments are discussed below (S1 Table). A heatmap of the additional 118 commonly downregulated transcripts along with their GO term designations are shown in S7 Fig.

The parasite's responses to KD of PfRSM22 or PfMRPL23 also included upregulation of many transcripts, including 177 commonly upregulated transcripts (Fig 7C). Interestingly, only 2 of the 177 upregulated transcripts were related to the parasite mitochondrion, including a putative ABC transporter (PF3D7_1352100; ATM-1 like) and a DNL-type Zinc finger protein (Pf3D7_1420300; HSP70-3 co-chaperone) (Fig 7B). Both genes are non-mutable as assessed by the large-scale mutagenesis study carried out using the PiggyBac transposon in *P. falciparum* [51]. A possibility exists that increased expression of these two proteins could be part of a feedback response to mitochondrial dysregulation. Apicoplast chaperone protein DnaJ (Pf3D7_0409400) was also slightly upregulated on day 6 following PfRSM22 KD and day 4 after PfMRPL23 KD (S6 Fig panel B). The upregulation of these transcripts as the parasite approaches its demise may indicate attempts to maintain a certain level of homeostasis. S8 Fig provides a heatmap of the remaining 175 commonly upregulated transcripts and related GO term pathways affected outside the parasite mitochondrion in both KD lines.

## Late effects of PfRSM22 and PfMRPL23 KD on transcripts of mitoribosomal components

The transcriptomic profiles related to mitoribosomal components at the late phase KD of PfRSM22 and PfMRPL23 were different from those seen at earlier times following KD. 28 out

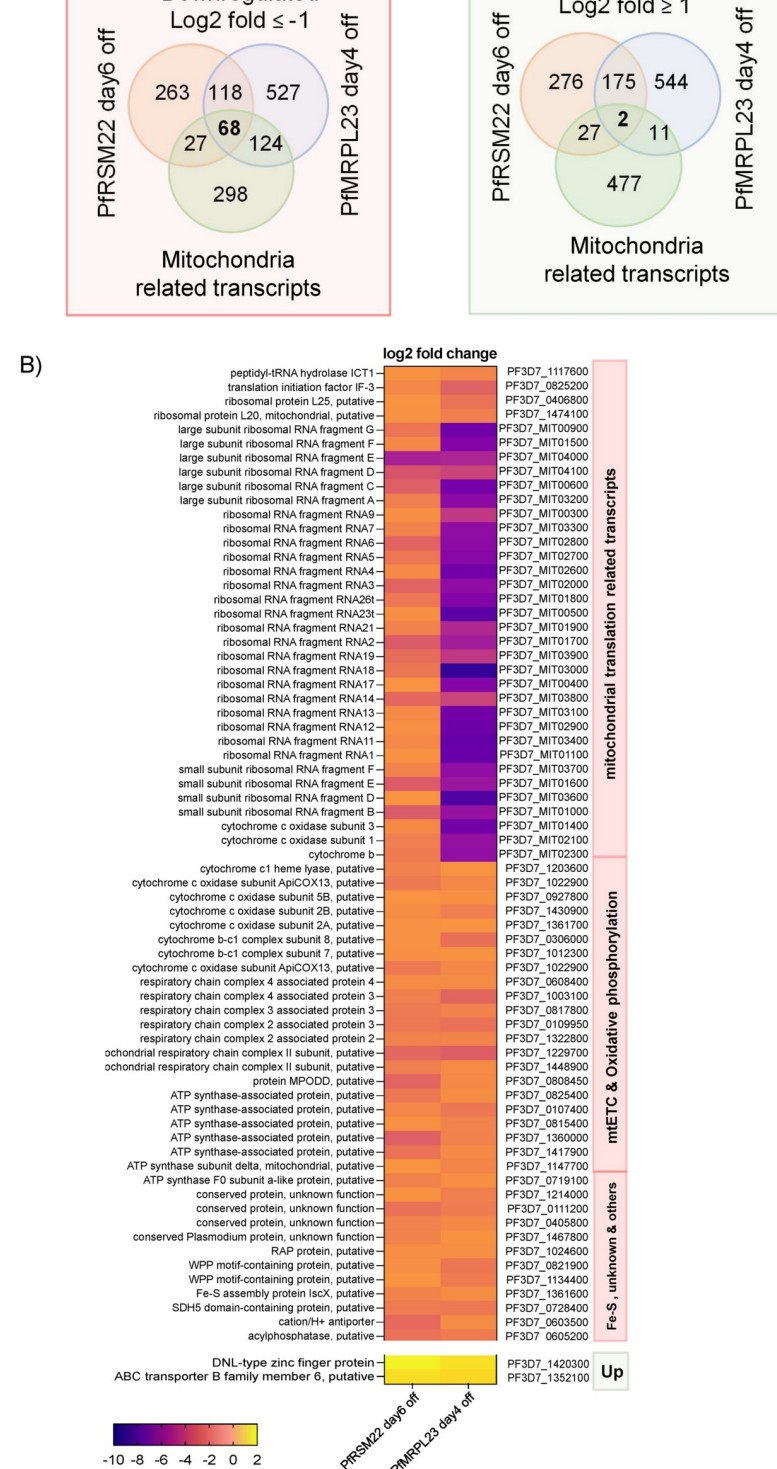

**Fig 7. Genes regulated in common after KD of PfRSM22 and PfMRPL23 in the late phase.** (A) Venn diagrams depict the relationship of downregulated gene transcripts in the PfRSM22 KD line on day 6 (orange) and in the PfMRPL23 KD line on day 4 (blue) to the 517 mitochondrial related transcripts (green). Transcripts downregulated ≥ 2-fold are included in this analysis. (B) Heat map of commonly downregulated (68 genes) and upregulated genes (2 genes) between PfRSM22 KD and PfMRPL23 KD at the late phase. (C) Venn diagrams depict the relationship of upregulated gene transcripts in the PfRSM22 KD line on day 6 (orange) and in the PfMRPL23 KD line

on day 4 (blue) to the 517 mitochondrial related transcripts (green). Transcripts upregulated $\geq$ 2-fold are included in this analysis.

of the 43 annotated PfMRPs were downregulated on day 4 following PfMRPL23 KD and only 3 SSU PfMRPs (S9, S11 and S12) were upregulated (Fig 8A, blue). The response of PfRSM22 KD on day 6 included downregulation of 13 PfMRPs (7 SSU and 6 LSU) (Fig 8A, orange). Yeast and mammalian orthologues of 9 out of these 13 PfMRPs belong to the categories of early and intermediate assembly proteins in the process of mitoribosomal formation [43]. Interestingly, only 3 PfAPRs (L9, L18, L21) were downregulated upon PfRSM22 KD and PfMRPL23 KD at later time points (S6 Fig panel B).

Among mt rRNAs, 34 out of 39 transcripts were downregulated following KD of PfRSM22 and PfMRPL23 at the late time points, except for a slight increase in transcription of RNA 15 (Pf3D7_MIT01200) upon PfRSM22 KD (Fig 8B). RNA 8 (Pf3D7_MIT04200), an mt SSU rRNA in domain II (S5 Fig), was exceptionally downregulated upon PfRSM22 KD. RT-qPCR results agree with the drastic reduction of RNA8 upon PfRSM22 KD in early and late phases of KD (S4 Fig Panel B). While the early phase of PfMRPL23 KD on day 2 had slight upregulation of 24 mt rRNA transcripts (Fig 5B), the late phase of PfMRPL23 KD caused a severe downregulation of almost all mtRNA transcripts. As shown in the early phase of KD, 5 mt RNA transcripts (RNA16, RNA20, SSUA, RNA27t, LSUB) remained undetected in the late phase KD in both parasite lines. Altogether, these data suggest collapse of mitoribosomes at the late phase KD of both PfRSM22 and PfMRPL23, which is consistent with malfunctioning of the parasite mitochondrion and other nuclearly encoded pathways before the onset of parasite demise.

## Potential novel *Plasmodium falciparum* MRPs

Studying mitoribosomes in *Plasmodium* is hindered by multiple biological and technical challenges of isolating relatively pure mitochondria in tractable quantity. As a result, we have a limited understanding of this divergent ribosome in *Plasmodium spp*. In addition, a large fraction of the *Plasmodium* proteome consists of proteins of unknown function, including 80 proteins present in our provisional mitochondrial proteome (S4 Table). Out of these 80 predicted mitochondrial conserved proteins of unknown functions, we found that 71 were differentially regulated upon PfRSM22 and/or PfMRPL23 KD (p < 0.05). Searching for potential roles of these mitochondrial proteins of unknown function, we performed structural modeling using Swiss model (https:// swissmodel.expasy.org/). Based on the modeling results, 16 of the differentially regulated mitochondrial proteins of unknown function exhibited structural similarity to known mitoribosomal proteins from other organisms. Transcripts for 13 out of these 16 proteins were significantly regulated following PfRSM22 and/or PfMRPL23 KD (S1 Table). These 13 proteins have a high isoelectric point (pI) due to the presence of basic amino acid residues. Eight out of 13 proteins of unknown function had predicted structural similarity to mitoribosomal proteins from another alveolate, *Tetrahymena thermophila* [32]. Two proteins were structurally similar to MRPs from *Trypanosoma brucei*, two were similar to plant MRPs and one to a human MRP. As expected, these potential PfMRPs listed in S1 Table lack orthologues in *Cryptosporidium parvum*, an apicomplexan that has lost mtDNA and, hence, lacks mitochondrial translation. Further studies would be needed to determine whether these proteins are divergent or lineage-specific *Plasmodium* MRPs.

## Conclusions

In this study, we have reported the importance of a putative mitoribosome assembly factor (PfRSM22) and a ribosomal protein of bacterial origin (PfMRPL23) in the asexual

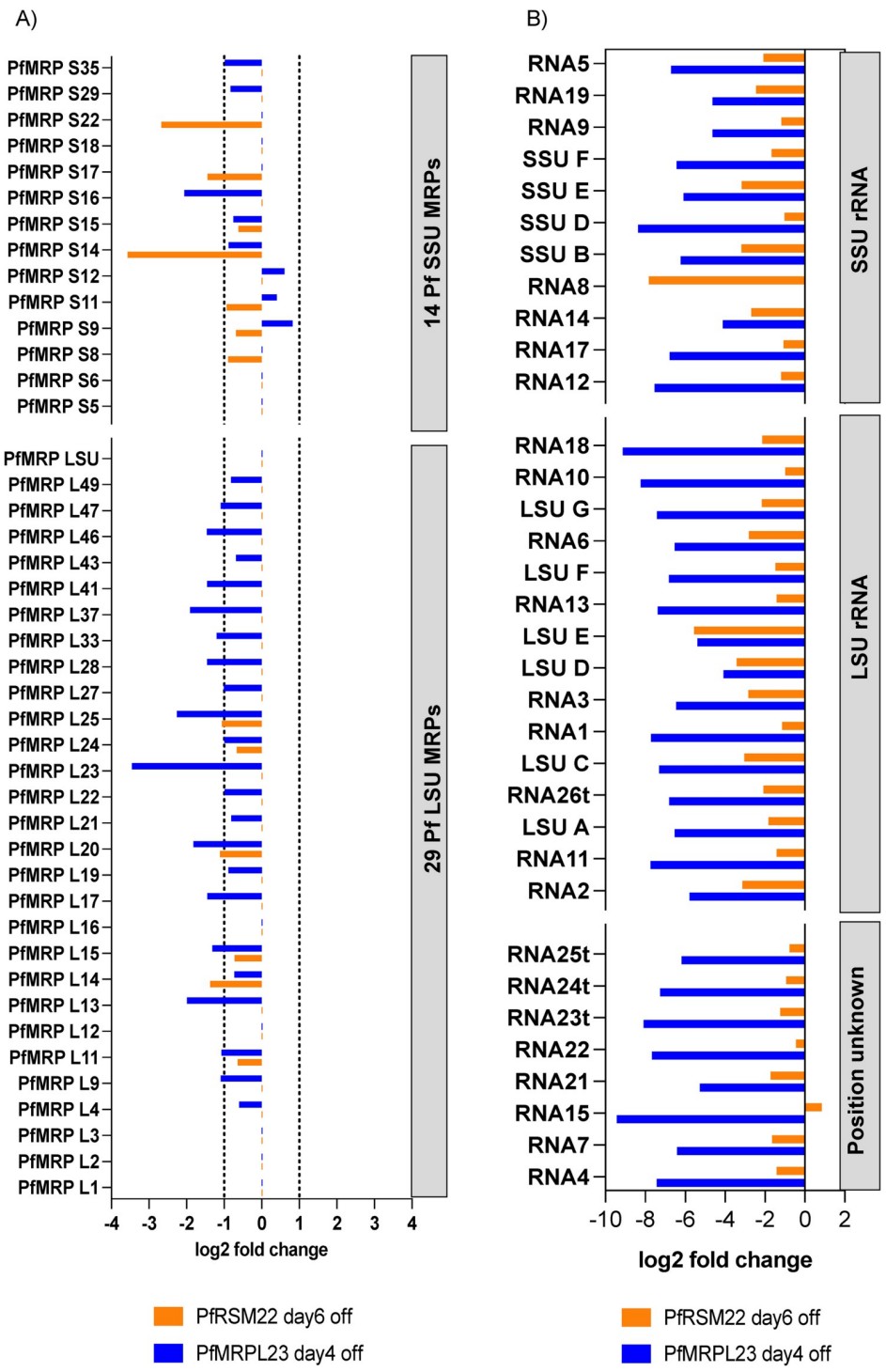

**Fig 8. Late effects of PfRSM22 and PfMRPL23 KDs on transcripts of mitoribosomal components.** (A) Log2 fold change of putative PfMRPs on day 6 after PfRSM22 KD (orange) and on day 4 after PfMRPL23 KD (blue). (B) Log2 fold change of mt rRNA fragments on day 6 after PfRSM22 KD (orange) and on day 4 after PfMRPL23 KD (blue).

development of *P. falciparum* malaria parasites. We used RNA sequencing as a tool to understand the parasite's responses in regulating gene expression upon KD of PfRSM22 and

PfMRPL23. To our knowledge, this is the first report highlighting a potential role of PfRSM22 in mitoribosomal assembly and affecting related downstream pathways leading to parasite death. PfMRPL23 KD on the other hand has a more immediate impact on parasites' metabolic pathways that is detrimental to the parasite one IDC sooner than PfRSM22 KD. The data also uncover global and mitochondrial specific transcriptomic changes over 6 days following PfRSM22 KD and 4 days following PfMRPL23 KD. In addition to that, we identified 13 conserved mitochondrial proteins of unknown function that are potential new MRPs in *P. falciparum*.

## Supporting information

**S1 Fig. PfRSM22 protein domain and amino acid sequence alignment.**
(DOCX)

**S2 Fig. PfMRPL23 protein domain and amino acid sequence alignment.**
(DOCX)

**S3 Fig. Endogenous tagging of PfRSM22 and PfMRPL23 using CRISPR/Cas9.**
(DOCX)

**S4 Fig. RT-qPCR of representative SSU and LSU mt rRNA upon protein KD.**
(DOCX)

**S5 Fig. Secondary structure of Pf mt rRNA fragments and their likely positions in the modeled SSU and LSU.**
(DOCX)

**S6 Fig. Early and late effects of PfRSM22 and PfMRPL23 KD on apicoplast related transcripts.**
(DOCX)

**S7 Fig. Downregulated non-mitochondrial transcripts common between PfRSM22 and PfMRPL23 KDs in the late phase.**
(DOCX)

**S8 Fig. Upregulated non-mitochondrial transcripts common between PfRSM22 and PfMRPL23 KDs in the late phase.**
(DOCX)

**S1 Table. List of primers and oligoes used in this study.**
(DOCX)

**S2 Table. Read count files of the PfRSM22_3HA line (aTc ON and day 2, day4, day 6 aTc OFF) and the PfMRPL23_3HA (aTc ON and day 2, day 4 aTc OFF).**
(XLSX)

**S3 Table. KEGG pathway and Gene ontology (GO) term list of significantly regulated genes upon KD of PfRSM22 and PfMRPL23.**
(XLSX)

**S4 Table. List of significantly up/downregulated genes, mitochondrial related genes, and genes in the Venn diagrams.**
(XLSX)

**S5 Table. RNA sequencing read depth.**
(XLSX)

**S1 File.**
(DOCX)

**S1 Raw images. Uncropped western blot images.**
(PDF)

## Acknowledgments

We are grateful for the constant support from members of the Center for Molecular Parasitology, Department of Microbiology and Immunology at Drexel University College of Medicine. We are thankful to Dr. Aishwarya Iyer (University of Alberta, Canada), Dr. Frank Bearof and Dr. Joshua Mell (Drexel University College of Medicine) for their guidance in RNA sequencing data analysis. We also thank Dr. Jacquin Niles (MIT) for the TetR-DOZI-aptamer plasmid, Dr. James Burns (Drexel University College of Medicine) for Exp2 antibody and Dr. Joshua Beck (Iowa State University) for the CRISPR/Cas9 plasmid. We thank vEuPathDB (https://veupathdb.org) for providing resources for data analysis.

## Author Contributions

**Conceptualization:** Swati Dass, Michael W. Mather, Liqin Ling, Akhil B. Vaidya, Hangjun Ke.

**Data curation:** Swati Dass, Michael W. Mather, Joanne M. Morrisey, Liqin Ling.

**Formal analysis:** Swati Dass, Michael W. Mather, Akhil B. Vaidya, Hangjun Ke.

**Funding acquisition:** Akhil B. Vaidya, Hangjun Ke.

**Investigation:** Swati Dass, Michael W. Mather, Joanne M. Morrisey, Liqin Ling, Hangjun Ke.

**Methodology:** Swati Dass, Michael W. Mather, Joanne M. Morrisey, Liqin Ling, Hangjun Ke.

**Project administration:** Swati Dass, Akhil B. Vaidya, Hangjun Ke.

**Resources:** Swati Dass, Hangjun Ke.

**Software:** Swati Dass, Michael W. Mather.

**Supervision:** Akhil B. Vaidya, Hangjun Ke.

**Validation:** Swati Dass, Michael W. Mather, Hangjun Ke.

**Writing – original draft:** Swati Dass, Hangjun Ke.

**Writing – review & editing:** Swati Dass, Michael W. Mather, Liqin Ling, Akhil B. Vaidya, Hangjun Ke.

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
