## [Decision Letter · Decision Letter 0]

1 Jun 2022

PONE-D-22-12422Transcriptional changes in Plasmodium falciparum upon conditional knock down of mitochondrial ribosomal proteins RSM22 and L23PLOS ONE

Dear Dr. Hangjun Ke,

Thank you for submitting your manuscript to PLOS ONE. After careful consideration, we feel that it has merit but does not fully meet PLOS ONE’s publication criteria as it currently stands. Therefore, we invite you to submit a revised version of the manuscript that addresses the points raised during the review process. ==============================

Your submission was analyzed by two external expert reviewers and ref#1 raised some serious problems and recommended rejection, while the problems raised by ref#2 were less serious and recommended minor revision. M

y editorial decision is "major revision" and when revising please pay particular attention to the comments of ref#1. My editior's comment is that in your revised manuscript to there absolutely has to be ENA reference numbers, so that reviewers can access and examine your RNAseq data.

We look forward to receiving your revised manuscript.

Kind regards,

Gordon Langsley

Academic Editor

PLOS ONE

Journal Requirements:

"The work was supported by NIH/NIAID grants to Dr. Hangjun Ke (K22AI127702) and Dr. Akhil Vaidya (AI028398)."

"This work was supported by an NIH/NIAID career transition award (K22AI127702) to Dr. Hangjun Ke and an NIH/NIAID R01 grant to Akhil B. Vaidya (AI028398). "

Reviewers' comments:

Reviewer's Responses to Questions

**Comments to the Author**

1. Is the manuscript technically sound, and do the data support the conclusions?

Reviewer #1: No

Reviewer #2: Yes

2. Has the statistical analysis been performed appropriately and rigorously? 

Reviewer #1: Yes

Reviewer #2: Yes

3. Have the authors made all data underlying the findings in their manuscript fully available?

Reviewer #1: No

Reviewer #2: Yes

4. Is the manuscript presented in an intelligible fashion and written in standard English?

Reviewer #1: Yes

Reviewer #2: Yes

5. Review Comments to the Author

Reviewer #1: In this manuscript, Dass et al. investigate the function of two putative components of the Plasmodium falciparum mitoribosome. Using conditional knock-downs, the authors convincingly show that both RSM23 and L23 are mitochondrially localized and are important fo parasite growth. They then go on to analyze RNA accumulation after 2 days of KD and after 4 and 6 days of KD (the latter only for PfMRPL23 KD). The study is rather descriptive and does not deliver insights into the molecular function of the two putative mitoribosomal proteins.

Major concerns

1. The authors make a number of conclusions based on mitochondrial RNA accumulation. This is problematic since RNA Seq is carried out using polyA-selected RNA. However, it is still an open question in the field, to what extend mitochondrial mRNA and rRNA are polyadenylated and what function polyadenylation has here. This may also be the reason for the failure to detect several known mitochondrial rRNA fragments. Thus, the bias introduced into the RNA seq data of mitochondrial transcripts is potentially huge. This is true if different time points are compared, but is even more relevant if comparisons between different genes are made. All comparisons between the accumulation of rRNA fragments from the small versus the large subunit are difficult. Therefore, it is essential to confirm conclusions on differential expression with orthogonal methods (qRT-PCR; RNA gel blot hybdirization).

2. Depleting mitoribosomes is expected to lead to massive problems with mitochondrial respiration, which in turn affects the cell on all levels. This obviously includes the accumulation of RNAs in all three genetic compartments. Thus, the RNA pattern observed by RNA seq is the secondary product of a number of problems these KD cells face and is not helpful to understand the primary function of the two proteins investigated. There are some interesting differences in RNA accumulation between the two knock-outs, which might be explained by the different severity of the mutants on mitochondrial translation, but in the end, no insights into the translational capacity of the mutants is shown. A targeted ribosome footprinting analysis or polysome analysis would be needed to understand the impact of the proteins on translation.

3. The two proteins investigated are presumed to be mitoribosome components based on sequence comparisons, but this has not been experimentally shown. In this light, the differential effects on the transcriptome might indicate different functions.

4. The idea that PfRSM22 is an assembly factor of the mitoribosome is not substantiated by any data. There is no evidence shown for “mitoribosome synthesis inhibition” (lines 146-148).

Minor concern

The authors do not provide access to the raw data which precludes any analysis on the reviewers’ site. Usually, SRA accessions (or the like) are made available for the reviewing process.

Reviewer #2: The authors present an article on the study of Plasmodium falciparum, of which two nuclear genes coding for mitochondrial proteins have been KD. Their choices fell on the PfRSM22 and PfMRPL23 genes without any reason being known (continuation of their previous studies ?). The authors used a conditional KD system to explore the transcriptomic bulk response of erythrocyte stages at two stages : early (day 2 post KD) and late (More than 6 days post KD) phases. By comparing the differential expression of other proteins, the authors propose a series of new potentially MRP-like proteins.

The manuscript is well written and very clear. The technical approach is convincing and the results are well presented. They attest to the localization of these two proteins in the mitochondria and also show that the KD of these genes leads to the death of the parasites at 8 and 6 days post KD for PfRSM22 and PfMRLP23 respectively.

However, I have two remarks :

The first concerns the methodology adopted in the analysis of the sequencing results. In their materials and methods the authors state that transcriptomic data are analyzed in duplicate using a set of tools including clusterProfiler R Package. The results presented are not exactly comparable with those found using PlasmoDB algorithm from Gene Ontology Enrichment pages using their avaliable data. It would seem relevant to me to discuss the convergences found between these two methodologies and how they could reinforce the robustness of the data presented.

The second remark concerns the relevance of analyzing the transcriptome at the late phase where it seems at least that a majority of the parasites are already dead (figure 1 E) and 1 F)), the data collected are therefore in my opinion difficult if not impossible to analyze and certainly subject to controversy.

6. PLOS authors have the option to publish the peer review history of their article (what does this mean?). If published, this will include your full peer review and any attached files.

Reviewer #1: No

Reviewer #2: No

---

## [Author Response · Author response to Decision Letter 0]

17 Aug 2022

Thank you for critical review of our manuscript. Please check the details of our responses to your comments in a separate word file entitled "point-to-point responses to review comments". Thanks.

---

## [Decision Letter · Decision Letter 1]

8 Sep 2022

Transcriptional changes in Plasmodium falciparum upon conditional knock down of mitochondrial ribosomal proteins RSM22 and L23

PONE-D-22-12422R1

Dear Dr. Hangjun Ke,,

We’re pleased to inform you that your manuscript has been judged scientifically suitable for publication and will be formally accepted for publication once it meets all outstanding technical requirements.

Kind regards,

Gordon Langsley

Academic Editor

PLOS ONE

Additional Editor Comments (optional):

Reviewers' comments:

Reviewer's Responses to Questions

**Comments to the Author**

1. If the authors have adequately addressed your comments raised in a previous round of review and you feel that this manuscript is now acceptable for publication, you may indicate that here to bypass the “Comments to the Author” section, enter your conflict of interest statement in the “Confidential to Editor” section, and submit your "Accept" recommendation.

Reviewer #2: All comments have been addressed

2. Is the manuscript technically sound, and do the data support the conclusions?

Reviewer #2: Yes

3. Has the statistical analysis been performed appropriately and rigorously? 

Reviewer #2: Yes

4. Have the authors made all data underlying the findings in their manuscript fully available?

Reviewer #2: Yes

5. Is the manuscript presented in an intelligible fashion and written in standard English?

Reviewer #2: Yes

6. Review Comments to the Author

Reviewer #2: (No Response)

7. PLOS authors have the option to publish the peer review history of their article (what does this mean?). If published, this will include your full peer review and any attached files.

Reviewer #2: No

---

## [Editor Report · Acceptance letter]

15 Sep 2022

PONE-D-22-12422R1 

Transcriptional changes in *Plasmodium falciparum* upon conditional knock down of mitochondrial ribosomal proteins RSM22 and L23 

Dear Dr. Ke:

I'm pleased to inform you that your manuscript has been deemed suitable for publication in PLOS ONE. Congratulations! Your manuscript is now with our production department. 

Kind regards, 

on behalf of

Dr. Gordon Langsley 

Academic Editor

PLOS ONE